# Retinal Microvascular Alterations in a Patient with Type 1 Diabetes Mellitus, Hemoglobin D Hemoglobinopathy, and High Myopia—Case Report and Review of the Literature

**DOI:** 10.3390/diagnostics13182934

**Published:** 2023-09-13

**Authors:** Alexandra Oltea Dan, Andrei Theodor Bălășoiu, Ileana Puiu, Andreea Cornelia Tănasie, Anca Elena Târtea, Veronica Sfredel

**Affiliations:** 1Department of Physiology, University of Medicine and Pharmacy of Craiova, 200349 Craiova, Romania; 2Department of Ophthalmology, University of Medicine and Pharmacy of Craiova, 200349 Craiova, Romania; 3Department of Pediatrics, University of Medicine and Pharmacy of Craiova, 200349 Craiova, Romania; 4Department of Neurology, University of Medicine and Pharmacy of Craiova, 200349 Craiova, Romania

**Keywords:** OCT angiography, diabetes type 1, hemoglobin D, high myopia

## Abstract

Type 1 diabetes mellitus (type 1 DM) is one of the most prevalent endocrinological diseases among children and young adults, with a growing incidence rate reaching up to 2.9 new cases per year per 100,000 persons below 15 years of age. We report a rare case of a 20-year-old female patient with type 1 DM, hemoglobin D (HbD) heterozygote variant and high myopia of −10.00 spheric diopters, and describe the retinal microvascular alterations visible on OCT angiography (angio-OCT). The patient also presented with a severe stature deficit (less than three standard deviations) and delayed puberty, which could not be explained only by suboptimal glycemic control and indicated possible hypopituitarism. HbA1c level evaluated with the high-performance liquid chromatography (HPLC) method was 6.5%, a falsely low value due to HbD hemoglobinopathy. On ophthalmic evaluation, the angio-OCT scan showed the following retinal microvascular alterations in the right eye (RE): the FAZ (Foveal Avascular Zone) area was 0.39 mm^2^, the FAZ perimeter was 2.88 mm, and the circularity index was 0.58. The following alterations were shown in the left eye (LE): the FAZ area was 0.34 mm^2^, the FAZ perimeter was 3.21 mm, and the circularity index was 0.41. Clinicians should consider high-performance retinal screening methods such as angio-OCT evaluation for young type 1 DM patients, especially for those with associated pathologies like high myopia and hemoglobinopathies. Moreover, multiple evaluation methods of HbA1c values are mandatory as hemoglobinopathies can interfere with the accuracy of HbA1c assay methods.

## 1. Introduction

Type 1 diabetes mellitus (type 1 DM) represents one of the most prevalent endocrinological diseases among children and young adults, with a growing incidence rate reaching up to 2.9 new cases per year per 100,000 persons below 15 years of age [1]. Although there is a continuous effort to improve insulin delivery systems and glucose monitoring systems, less than a third of type 1 DM patients achieve optimal glycemic control [2]. The major risk factors for developing retinal microvascular complications are poor metabolic control and the duration of the disease. Diabetic retinopathy (DR) represents the most frequent diabetic microvascular complication, and it develops progressively with the duration of the disease, potentially leading to proliferative diabetic retinopathy (PDR) and irreversible loss of vision. The prevalence of PDR increases to 25% after 15 years’ duration of the disease [3]. Therefore, early diagnosis of DR has an enormous therapeutic potential in preventing or delaying vision-threatening complications in young type 1 DM patients.

High myopia is characterized by axial elongation, stretching of the posterior eye wall, and retinal thinning, which can lead to multiple complications such as retinal detachment, macular holes, choroidal neovascularization, and retinoschisis [4].

The presence of both diabetes and high myopia in the same patients is associated with higher retinal nerve fiber layer (RNFL) damage than either pathology alone [5]. The mechanical stretching of the myopic eye wall and diabetic neural degeneration accelerate the thinning of the inner retinal layer [6], acting as cumulated risk factors for the development of retinal complications. The mechanisms of retinal neuronal degeneration in diabetes lead to the death of neuronal cells in the retina, mainly through glial activation and apoptosis. Hyperglycemia triggers the activation of protein kinase C (PKC), oxidative stress, and glial activation. The loss of retinal ganglion cells (RGC) is accelerated by high glial activation, proliferation, phagocytosis, the production of toxins, and finally the induction of RGC apoptosis.

Hemoglobin (Hb) is the main protein component of erythrocytes, mediating the transport of oxygen and carbon dioxide in the blood [7]. Adult hemoglobin A (HbA) is present in 95% of the adult population and is a tetrameric protein (α2β2) consisting of two α and two β subunits held together by noncovalent interactions [8]. Hemoglobinopathies are highly heterogeneous, monogenic disorders related to hemoglobin synthesis [9]. Hemoglobin D (HbD) is a β chain hemoglobin variant which was first described in 1951. It is also known as HbD Punjab or HbD Los Angeles and it is the fourth most frequently occurring hemoglobinopathy [10]. HbD differs in structure from HbA, at position 121 of the β chain where glutamine replaces glutamic acid [11]. Specific hemoglobin variants such as HbD may interfere with the accuracy of HbA1c values depending on the evaluation method used [12]. We report a case of a type 1 DM patient with HbD hemoglobinopathy and high myopia and discuss the retinal microvascular alterations of these conditions, visible on an OCT angiography (angio-OCT) evaluation.

## 2. Materials and Methods

### Case Presentation

We present a rare case of a 20-year-old female patient with type 1 DM, HbD heterozygote variant, high myopia, and microvascular retinal alterations found on angio-OCT. This research was approved by the Ethics Committee of the University of Medicine and Pharmacy of Craiova (project identification code 8612/07/06/2021) and was carried out in accordance with the rules of the Declaration of Helsinki, revised in 2013. Written informed consent was obtained from the patient.

Her medical history revealed that the patient was an only child, had a birth weight of 2800 g, and no history of prematurity. The patient was diagnosed with type 1 DM at the age of two and has been on basal-bolus insulin therapy since the diagnosis. While monitoring HbA1c, she was also diagnosed with HbD hemoglobinopathy. During childhood, severe stature deficit (less than 3 standard deviations) and delayed puberty were observed, which could not be explained only by suboptimal glycemic control. These clinical manifestations indicated a possible hypopituitarism, characterized by severe growth deficiency and pubertal delay, without the characteristic signs of dysmorphic syndromes. The patient’s family did not consent to growth hormone investigations and possible treatment for hypostature. Her family history revealed that her parents were of relatively normal height, the mother’s height being 162 cm and her father’s height being 165 cm.

At the age of 7, the patient was also diagnosed with mild myopia of −2.00 spheric diopters (SD) in both eyes (BE) which progressed up to −10.00 SD currently, at the age of 20. The patient attended yearly ophthalmic examinations after the age of 7, screening for diabetic retinopathy was done by dilated fundus examination, which was unremarkable, but there was a progressive increase in myopia, leading to high myopia. However, OCTA retinal scans were taken for the first time during her most recent ophthalmic examination.

Her mother was also diagnosed with myopia in adolescence and has a current refractive error of −2.50 SD BE. No family members have been diagnosed with diabetes mellitus and they declined further tests for hemoglobinopathy.

The patient’s current height is 149 cm (less than 3 standard deviations) and her weight is 45 kg, with a body mass index (BMI) of 20.3 kg/m^2^. For the past 2 years, she has been on a basal-bolus insulin regimen of 3 fixed doses of Aspart insulin (5 units before breakfast, 10 units before lunch, and 5 units before dinner) and 20 units of glargine insulin in the evening. With this therapy, her most recent HbA1c value was 6.5%, measured with high-performance liquid chromatography (HPLC) (Table 1). However, the average glycemic levels from her glucometer for the past 3 months varied between 160 and 184 mg%, which corresponds to a HbA1c value between 7 and 8%. This discrepancy is due to the presence of HbD identified in her blood. Her most recent hemoglobin electrophoresis (EDTA blood/capillary electrophoresis) revealed a percentage of HbA of 56.6% and abnormal HbD fractions of 40.4%. Complete blood count, serum electrolytes, liver transaminases, kidney function tests, and lipid profile were within normal parameters (Table 1).

Ophthalmic examination revealed a best-corrected visual acuity (BCVA) of 0.7 LogMar BE with a refractive error of −10.00 SD BE. Intraocular pressure measured using an ICare tonometer was 17 mmHg in the right eye (RE) and 18 mmHg in the left eye (LE). Slit lamp examination revealed normal anterior segment in BE, with no relative afferent pupillary defect.

Dilated fundus examination of BE revealed mild peripapillary atrophy, a thin retina with visible choroidal vasculature, and no clinical signs of DR (Figure 1).

Macular OCT examination revealed mild retinal thinning in the macular area, with a retinal thickness of RE 212 µm and LE 220 µm, without other alterations specific to high myopia or DR. Optic disc OCT examination showed a mean cup-to-disc (C/D) ratio of 0.3, with normal retinal nerve fiber layer thickness in BE.

Angio-OCT scans (Figure 2) revealed an enlarged foveal avascular zone (FAZ) area and perimeter, with a decreased circularity index and a decrease in capillary density of the superficial and deep capillary plexus in BE. In the RE, the FAZ area was 0.39 mm^2^, the FAZ perimeter was 2.88 mm, and the circularity index was 0.58; in the LE, the FAZ area was 0.34 mm^2^, the FAZ perimeter was 3.21 mm, and the circularity index was 0.41. Although the patient had previously attended regular ophthalmic examinations for DR screening, this was the patient’s first retinal angio-OCT evaluation.

## 3. Discussion

### Review of the Literature

Until now, only a few studies have focused on the coexistence of type 1 DM and HbD hemoglobinopathy in the same patients and the way that HbA1c evaluation methods can be influenced by the HbD variant. We searched the PubMed database for the following keywords: hemoglobin D, diabetes mellitus, and HbA1c. We found a total of five scientific articles, two original research studies and two case reports (Table 2); a full-text evaluation was performed.

Little RR and collaborators [13] demonstrated that the presence of both HbD and hemoglobin E variants interfered with ion-exchange HPLC methods of assessment of HbA1c, producing artificially low results for HbA1c. In their research regarding the interference of HbD on measurements of HbA1C, Lorenzo Medina and collaborators [14] showed that in diabetic patients who were heterozygous for the HbD variant, HbA1c levels using the ADAMS HA-8160 HPLC method gave falsely low or unquantifiable results due to an abnormal separation of HbA1c. For HbD-variant patients, they recommended the measurement of HbA1c using other alternatives, such as a turbidimetric immunoassay, which is less likely to be subject to interference. Pinés Corrales PJ and collaborators [15] presented a case report of the interference of HbD on the measurement of HbA1c, indicating that HPLC assay for HbA1c indicated lower values compared to those of serum glucose. Copplestone S and collaborators [16] reported a case of a patient with poorly controlled diabetes but apparently normal levels of HbA1C assessed with the HPLC method. Shukla A and collaborators [17] were the only ones reporting a case where HbA1c values were falsely elevated when using the HPLC method of assay. 

HbA1c represents a minor fraction of adult hemoglobin, which is formed slowly and nonenzymatically from hemoglobin and glucose [18]. Erythrocytes are freely permeable to glucose, therefore HbA1c is formed throughout their lifespan of approximately 120 days. The rate of HbA1c formation is directly proportional to ambient glucose concentration. Monitoring HbA1c is indispensable for the treatment and follow-up of type 1 DM patients as it reflects their glycemic control in the previous 8–12 weeks [19]. There are several factors that can interfere with HbA1C assay and may lead to less accurate results, and one of these factors is represented by hemoglobin variants, also called hemoglobinopathies. Multiple methods for measuring HbA1c levels are available, including cation-exchange HPLC, immunoassays, and enzyme-based assays. The most used method for measuring HbA1c levels is cation-exchange HPLC due to its fast and reliable results, but this method can be vulnerable to the effects of hemoglobin variants [20]. Currently, more than 1000 Hb variants have been identified, with most of them being clinically silent. 

Metabolic control and duration of type 1 DM are major factors influencing the risk of retinal microvascular complications leading to DR [21]. Our patient was diagnosed with type 1 DM at the early age of two, and shortly afterwards, she was diagnosed with HbD heterozygote variant. She was also diagnosed with myopia of −2.00 SD at the age of seven, progressing to −10.00 SD at 20 years old. She has a current duration of type 1 DM of 18 years, which increases the risk of DR complications. On ophthalmic examination, she presented with microvascular retinal alterations visible only on angio-OCT, namely enlarged FAZ area and perimeter, with a decreased circularity index in BE. These retinal alterations can be attributed to both myopia and type 1 DM and can accelerate the development of sight-threatening complications. Normal FAZ in healthy subjects has a circular or slightly elliptical shape which becomes increasingly irregular in patients with DR, altering the FAZ perimeter and FAZ area as well [22]. Although the patient’s kidney function tests were within the normal parameters, diabetic microvascular retinal changes and the preclinical morphologic changes of diabetic nephropathy are closely associated. Therefore, OCTA retinal monitoring of type 1 DM and hemoglobinopathy patients represents a reliable method to detect early retinal blood flow alterations, which may also reflect similar microvascular changes occurring elsewhere in the body, such as renal glomerulus, and may lead to a different approach in the overall clinical management of these patients.

Myopia has a multifactorial etiology determined by a complex interaction of genetic and environmental factors such as higher socioeconomic status, excessive near-work activity, and lack of outdoor activity [23]. Several research studies have investigated the retinal parameters of patients with myopia and diabetes mellitus, with variable results. Some results suggest that hyperglycemia may predispose one to juvenile-onset myopia [24] through higher levels of free insulin-like growth factor (IGF)-1 and lower levels of IGF-binding protein 3, leading to unregulated scleral growth and myopia. Ying Xiao and collaborators investigated whether the progression of myopia is accelerated in children with type 1 DM, and they found evidence that myopia progression is accelerated in type 1 DM children compared to non-diabetic children [25]. Moreover, angio-OCT studies of patients with high myopia and diabetes type 2 (T2D) found that parafoveal vascular density was decreased in myopic T2D patients compared with those without myopia [26].

Although the technology used in the treatment of type 1 DM is constantly advancing, aiming towards better glycemic control [27], other sight-threatening pathologies of patients must be taken into consideration when monitoring the possible retinal complications. High-resolution OCT techniques which allow for the evaluation of retinal layers [28] along with angio-OCT microvascular retinal evaluation are necessary screening tools for preventing sight-threatening complications in young diabetic patients with high myopia.

## 4. Conclusions

When monitoring the metabolic control of type 1 DM patients with known haemoglobinopathies, such as HbD, HbA1c levels measured with HPLC assay might fail to reflect the real mean values for glycemia. Awareness of Hb variants affecting HbA1c measurements is essential to avoiding the poor management of diabetic patients, because haemoglobinopathies can produce an artifact whereby the hemoglobin variant is being measured instead of or in addition to HbA1c. 

As the incidence of Hb variants is continuously increasing due to immigration, misleading HbA1C levels are becoming more frequent and should be tested with multiple methods, such as turbidimetric immunoassay, or different tests, such as glycated serum protein or glycated albumin, should be considered.

In the rare coincidence of type 1 DM, HbD hemoglobinopathy, and high myopia, advanced retinal screening methods such as angio-OCT are essential in order to identify early microvascular retinal alterations. Moreover, close monitoring of patients’ metabolic control through multiple HbA1c evaluation methods is mandatory, as the HPLC assay method underestimates real HbA1c values in patients with HbD hemoglobinopathy.

## Figures and Tables

**Figure 1 diagnostics-13-02934-f001:**
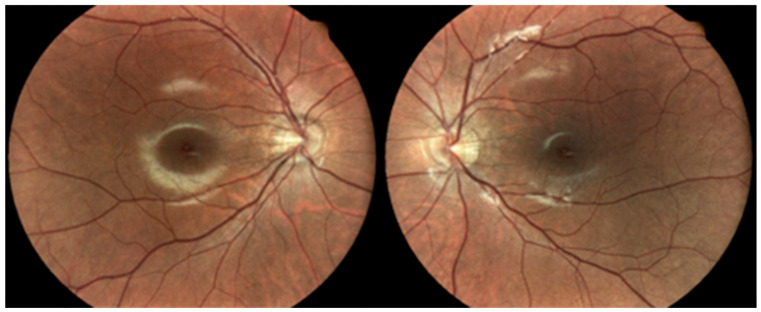
Dilated fundus photos showing mild peripapillary atrophy in BE, thin retina with visible choroidal vasculature, and no clinical signs of DR.

**Figure 2 diagnostics-13-02934-f002:**
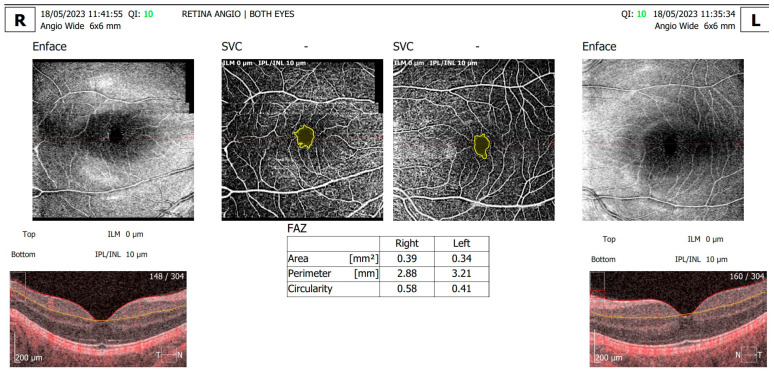
Angio-OCT measurements of a wide area of 6 × 6 mm around the fovea, showing alterations of FAZ parameters in BE, R—right eye, L—left eye.

**Table 1 diagnostics-13-02934-t001:** Biochemical investigations.

Laboratory Test (Units)	Results	Reference Range
C-reactive protein (CRP) (mg/dL)	0.09	<0.5
Alanine transaminase (ALT) (IU/L)	11.1	<33
Aspartate aminotransferase (AST) (IU/L)	14.6	<32
Total bilirubin (mg/dL)	0.39	≤1
Direct bilirubin (mg/dL)	0.14	≤0.3
Indirect bilirubin (mg/dL)	0.25	≤1
Glycated hemoglobin (Hb A1c) (%)	6.5	≥6.5 diabetes mellitus
Rheumatoid factor (UI/mL)	<10	<14
Serum iron (µg/dL)	51.2	33–193
Urine albumin (mg/L)	8.1	
Urine creatinine (mg/dL)	136.4	
Urine albumin–creatinine ratio (uACR) (mg/g)	5.93	<30
Hemoglobin electrophoresis (EDTA blood/capillary electrophoresis)		
Hemoglobin A (%)	56.6	96.7–97.8
Hemoglobin A2 (%)	3	2.2–3.2
Abnormal hemoglobin fractions *	40.4	
Hemoglobin (g/dL)	12.7	11.7–15.5
White blood cells (WBC) (10^3^/µL)	8.97	4–10
Red blood cells (RBC) (10^6^/µL)	4.54	3.8–5.1
Hematocrit (%)	37	35–45
Platelets (10^3^/µL)	358	150–450
Thrombin time (s)	12	9.8–12.1
INR	1.04	
Partial thromboplastin time (s)	30.9	21.6–28.7
Ionized calcium (mg/dL)	4.04	3.82–4.82
Serum calcium (mg/dL)	9.24	8.6–10
Total serum protein (g/dL)	7.14	6.6–8.7
HDL cholesterol (mg/dL)	57.2	≥60
LDL cholesterol (mg/dL)	89.9	<100
Triglyceride (mg/dL)	12.7	<30
Total cholesterol (mg/dL)	151.2	<200
Serum lipids (mg/dL)	493.7	400–700
Serum creatinine (mg/dL)	0.61	<1.1
Glomerular filtration rate (ml/min/1.73 m^2^)	130	≥90
Urine white blood cells (/µL)	100	negative
Urine protein (mg/dL)	negative	negative
Urine glucose (mg/dL)	300	negative

* On capillary hemoglobin electrophoresis, an abnormal fraction of hemoglobin (40.4 percent of the total hemoglobin) migrated into zone D, corresponding to heterozygous for HbD status.

**Table 2 diagnostics-13-02934-t002:** Research articles from the PubMed database for the following keywords: hemoglobin D, diabetes mellitus, and HbA1c.

Description of Results	Article Title	Type of Study/Year Published	Authors
Ion-exchange HPLC methods showed interference of HbD traits, producing artificially low HbA1c results.	Effects of hemoglobin (Hb) E and HbD traits on measurements of glycated Hb (HbA1c) by 23 methods [13]	Original research paper/ 2008	Little RR, Rohlfing CL, Hanson S, Connolly S, Higgins T, Weykamp CW, D’Costa M, Luzzi V, Owen WE, Roberts WL [13]
In 16 patients, the HPLC method indicated “abnormal separation”, and no result was given for HbA1c. For the remaining 11 patients, HbA1c results were abnormally low compared to the patients’ respective fasting glucose concentrations.	Interference of Hemoglobin D on Measurements of Hemoglobin A1c by the High-Performance Liquid Chromatography HA-8160 in 27 Patients [14]	Original research paper/ 2012	Lorenzo-Medina M, De-La-Iglesia S, Ropero P, Martin-Alfaro R, Quintana-Hidalgo L [14]
Patient with known type 2 diabetes for over 25 years treated with insulin for over 15 years was heterozygous for HbD Los Angeles. Using HPLC for HbA1c assay, HbA1c levels were low compared to those of serum glucose.	Interference of Hb D-Los Angeles on the measurement of glycated hemoglobin. A case report [15]	Case report/ 2017	Pinés Corrales PJ, Martínez López R, González Cabrera A, Ibáñez Navarro P, Vicente Albiñana Á [15]
The HbD variant reduced the apparent value of total HbA1c, when assessed using the HPLC method.	Normal glycated haemoglobin in a patient with poorly controlled diabetes mellitus and haemoglobin D Punjab: implications for assessment of control [16]	Case report/ 2002	Copplestone S, Mackay R, Brennan S [16]
In this case, due to interference of HbD, the results obtained for HbA1c using HPLC were falsely elevated.	Interference of hemoglobin D Punjab on measurements of glycated hemoglobin	Case report/ 2015	Shukla A, Dabadghao S, Gupta S, Verma S [17]

## Data Availability

The authors declare that the data for this research are available from the correspondence authors upon reasonable request.

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
