# Peer review of "Retinal Microvascular Alterations in a Patient with Type 1 Diabetes Mellitus, Hemoglobin D Hemoglobinopathy, and High Myopia—Case Report and Review of the Literature"

_diagnostics, 2023, doi:10.3390/diagnostics13182934_

Round 1
Reviewer 1 Report
The authors reported a rare case of a young adult with type 1 diabetes and Hemoglobin D hemoglobinopathy. The authors also reviewed current literature on how Hemoglobinopathy affects A1C measurements. The presence of hemoglobinopathy leads to an artificially lower value of A1C, resulting in suboptimal control of glycemia and consequently altering microvasculature in the eyes. I only have a few minor comments:
1. How did this combination of diabetes and hemoglobinopathy affect microvasculature in other organs, such as the kidney?
2. Was there longitudinally data to monitor disease progress in this patient?
3. What about family history of myopia, diabetes, and hemoglobinopathy? Is it possible that the patient may carry certain genetic mutations that lead to these symptoms/conditions?
Author Response
Dear Reviewer,
Thank you for helpful comments. Please see the attachment.

Reviewer 2 Report
The manuscript entitled "Retinal microvascular alterations in a patient with type 1 diabetes, Hemoglobin D hemoglobinopathy and high myopia – case report and review of literature" brings an interesting case report about retinal vascular complications in a patient with Type 1 DM and high myopia.
Observations
Abstract
Please explain FAZ before abbreviation.
Introduction - please replace Type 1 diabetes with type 1 diabetes mellitus (type 1 DM).
Please explain in more details the neuronal degeneration mechanism in type 1 DM - line 53-55.
Please write the Ethic Committee Approval number and mention if the patient signed an informed consent about the data you are presenting in this manuscript.
Line 106 - please mention the method of intraocular pressure measurement.
Table 2 - please organize the table starting with the description of the results in the first column and with the authors in the last column.
Please explain the abbreviation for BE, RE, Le before using it (both eye, right eye and left eye respectively).
Author Response
Dear reviewer,
Thank you for your helpful comments. Please see the attachment.
